# Effects of the Biofertilizer OYK (*Bacillus* sp.) Inoculation on Endophytic Microbial Community in Sweet Potato

**Ahsanul Salehin [1], Md Hafizur Rahman Hafiz [2,3], Shohei Hayashi [2], Fumihiko Adachi [2] and Kazuhito Itoh [1,2,*]**

[1] The United Graduate School of Agricultural Sciences, Tottori University, 4-101 Koyama-minami, Tottori 680-8553, Japan; ujanrijvi224@gmail.com

[2] Faculty of Life and Environmental Sciences, Shimane University, 1060 Nishikawatsu, Matsue, Shimane 690-8504, Japan; hafizhstu@hotmail.com (M.H.R.H.); shohaya@life.shimane-u.ac.jp (S.H.); fadachi@life.shimane-u.ac.jp (F.A.)

[3] Department of Crop Physiology and Ecology, Hajee Mohammad Danesh Science and Technology University, Dinajpur 5200, Bangladesh

[*] Correspondence: itohkz@life.shimane-u.ac.jp; Tel.: +81-852-32-6521

**Abstract:** Sweet potato (*Ipomoea batatas* L.) grows well even in infertile and nitrogen-limited fields, and endophytic bacterial communities have been proposed to be responsible for this ability. Plant-growth-promoting bacteria are considered eco-friendly and are used in agriculture, but their application can interact with endophytic communities in many ways. In this study, a commercial biofertilizer, OYK, consisting of a *Bacillus* sp., was applied to two cultivars of sweet potato, and the effects on indigenous endophytic bacterial communities in field conditions were examined. A total of 101 bacteria belonging to 25 genera in 9 classes were isolated. Although the inoculated OYK was not detected and significant plant-growth-promoting effects were not observed, the inoculation changed the endophytic bacterial composition, and the changes differed between the cultivars, as follows: *Novosphingobium* in α-Proteobacteria was dominant; it remained dominant in Beniharuka after the inoculation of OYK, while it disappeared in Beniazuma, with an increase in *Sphingomonas* and *Sphingobium* in α-Proteobacteria as well as *Chryseobacterium* and *Acinetobacter* in Flavobacteria. The behavior of Bacilli and Actinobacteria also differed between the cultivars. The Shannon diversity index (*H*) increased after inoculation in all conditions, and the values were similar between the cultivars. Competition of the inoculant with indigenous rhizobacteria and endophytes may determine the fates of the inoculant and the endophytic community.

**Keywords:** OYK; biofertilizers; PGPR; sweet potato; endophytes; microbial community; shannon diversity index

## 1. Introduction

Modern agriculture systems are being intensified through the use of various technologies to achieve maximum efficiency and high qualify products to meet the growing global demand for food supply [1]. At present, as a part of agricultural intensification, crop production depends on the large-scale use of chemical fertilizers [2]. However, the intensive use of chemical fertilizers can result in considerable negative environmental impacts and pollution [3]. Therefore, an alternative strategy is urgently needed to establish sustainable agriculture and ecological balance in agro-ecosystems.

Plant-growth-promoting rhizobacteria (PGPR) are free-living soil bacteria that enhance plant growth by colonizing the rhizosphere [4]. PGPR regulate nutritional and hormonal balance, produce

phytohormones, solubilize nutrients, and induce resistance to plant pathogens [5]. Therefore, PGPR have been used as biofertilizers and/or bioenhancers, as an alternative source of chemical fertilizers to improve soil quality and sustainability and to increase crop production [6–8]. The application of PGPR has become a more broadly recognized practice for the enrichment of sustainable agricultural production in several parts of the world.

Sweet potato (*Ipomoea batatas* L.) is a resilient, easily propagated crop, and its roots are largely used for food consumption. More than 95% of the global sweet potato crop is produced in developing countries, and it has vast economic and social importance [9,10]. It is also well-known for its ability to grow well even in infertile and nitrogen-limited fields [11,12], and nitrogen fixation by endophytic bacteria has been proposed to contribute to this attribute [13].

Endophytes are known to promote plant growth by producing phytohormones [14–16] and siderophores [17,18] and through nitrogen fixation [19]. It has also been reported that some endophytes can protect plants by producing antipathogenic substances [20], ameliorating disease development [21], and inducing stress tolerance [22]. Therefore, an understanding of the endophyte–plant interaction is essential for developing sustainable systems of crop production [23].

Diverse endophytic bacteria have been isolated from sweet potato; such bacteria include *Gluconacetobacter*, *Klebsiella*, and *Pantoea* [24,25], as well as *Enterobacter*, *Rahnella*, *Rhodanobacter*, *Pseudomonas*, *Stenotrophomonas*, *Xanthomonas*, and *Phyllobacterium* [26]. Marques et al. [27] and Puri et al. [28] reported 93 and 243 endophytic bacterial strains belonging to 17 and 34 genera in Brazilian and Nepalese sweet potatoes, respectively. Among these isolates of sweet potato bacterial endophytes, many strains had beneficial properties, such as nitrogen fixation, auxin production, antagonistic effects, phosphate solubilization, and siderophore production.

It is speculated that the beneficial functions of endophytes are realized when a suitable endophytic community is established, and it is expected that the inoculation of PGPR has synergic or competitive effects on the composition and function of the endophytic community [29]. However, to the best of our knowledge, only a few studies are available on this subject. Conn and Franco [30] showed that the inoculation of a nonadapted microbial inoculum into the soil disrupted the natural actinobacterial endophyte population of wheat plants and reduced their diversities and colonization levels, whereas the inoculation of a single actinobacterial endophyte did not affect the indigenous endophyte population. Gadhave et al. [31] reported that seed and soil inoculations of *Bacillus* spp. changed the composition of the endophytic bacterial community of sprouting broccoli and increased its diversity, as established through the metagenomic approach.

In the present study, we treated sweet potato with a commercial biofertilizer, OYK, consisting of a *Bacillus* strain, which was reported to induce plant tolerance to abiotic and/or biotic stresses and to have antimicrobial activities against pathogens [32]. We then examined culturable endophytic communities at harvest in order to obtain further information on the effects of PGPR inoculation on indigenous endophytic bacterial communities in field conditions.

## 2. Materials and Methods

### 2.1. Growth Condition, Inoculation, and Cultivation of Sweet Potato

Two cultivars of sweet potato, Beniazuma (A) and Beniharuka (H), were used in this study. OYK Farming Ace (Hamaguchi Institute of Microbiology Inc., Kyoto, Japan, http://www.oyk.jp/), consisting of about 8E + 9 CFU/mL endospores of one *Bacillus* sp. strain (LC590219), was used as PGPR, according to the manufacturer's instruction. One milliliter of OYK solution was diluted to 4 L with sterilized distilled water, and twelve seedlings of each cultivar were dipped in the solution for 60 h (O). The same numbers of the seedlings were soaked in distilled water as a control (C). These seedlings were transplanted at random at 20 cm intervals on ridges with 1 m spacing in a rooftop experimental field [33] at Shimane University in Shimane, Japan. The field was filled with artificial soil (Viva soil; Toho Leo Co., Osaka, Japan) that had high porosity (45%) and contained very little nutrition, and a

chemical fertilizer (N:$P_2O_5$:$K_2O$ = 4:8:15 g/m$^2$) was applied before planting. The plants were cultivated from June to November in 2015 with drip irrigation (Super Typhoon NETAFIM Co., Tel Aviv, Israel).

## 2.2. Sample Collection and Isolation of Endophytic Bacteria

At harvest, the fresh weights of the shoots and tubers of each sweet potato plant were measured. Culturable endophytes of sweet potato tubers were examined; among the plant parts, the highest population was observed in tubers in our previous study [34]. The surface of each tuber sample was washed with running tap water for 10 min and cut longitudinally with a sterilized knife at its middle part after wiping off the water with a paper towel. Then, the cut surface was stamped on modified MR agar medium, with and without the supplementation of ammonium nitrate as a nitrogen source [35] in a petri dish. The ingredients of the media are listed in Supplementary materials Table S1. The efficiency of the washing procedure was evaluated by stamping the surface of the washed tubers on agar media. After incubation for 2 days at ca. 26 °C, all the bacterial colonies were transferred to either N-supplemented or N-free MR media for purification and then grouped based on their morphologies on the two media. Based on their relative abundance, 1–3 representative isolates from each group, comprising 30–81% of total isolates, were selected for further analysis (Table 1).

**Table 1.** Number of isolated endophytic bacterial strains of sweet potato, types of morphologies on the agar plates, and strains selected for sequence analysis.

| Sample [a] | CFU [b] | Isolated [c] | Morphology [d] | Selected [e] | Identified [f] |
|---|---|---|---|---|---|
| AO-N(+) | 32 | 32 | 11 | 14 | 14 |
| AO-N(−) | 42 | 40 | 17 | 17 | 17 |
| AC-N(+) | 22 | 13 | 6 | 10 | 10 |
| AC-N(−) | 24 | 18 | 9 | 12 | 11 |
| HO-N(+) | 50 | 50 | 12 | 15 | 13 |
| HO-N(−) | 46 | 42 | 14 | 15 | 12 |
| HC-N(+) | 31 | 21 | 11 | 13 | 13 |
| HC-N(−) | 22 | 16 | 11 | 13 | 11 |
| Total | 269 | 232 | - | 109 | 101 |

[a] Endophytic strains were isolated from the sweet potato cultivars, Beniazuma (A) and Beniharuka (H). Sweet potato seedlings were inoculated with OYK (O) as PGPR or with distilled water as the control (C). The modified MR agar medium was used for isolation, with nitrogen supplementation (N(+)) or without a nitrogen (N(−)) source. [b] Number of colonies that appeared on the original agar plates. [c] Number of successfully isolated colonies. [d] Number of morphologies observed. [e] Number of isolates selected based on the relative abundances of morphologies for sequence analysis. [f] Number of strains successfully sequenced.

## 2.3. Genetic Analysis of Endophytes

Genomic DNA was extracted from each isolate, as described by Saeki et al. [36], with slight modifications, and used as a template for PCR for the amplification of the partial 16S rRNA gene sequence. As an indication of the dinitrogen-fixing potential of the isolates, *nifH* genes, which encode nitrogenase reductase, were PCR-amplified, for which a small amount of culture was directly used as a template. The primers used were fD1 and rP2 [37] and PolF and PolR [38] for the 16S rRNA and *nifH* genes, respectively. The components of the PCR master mixtures and the PCR running conditions are summarized in Supplementary materials Table S2. PCR products were purified and subjected to PCR cycle sequencing, according to the procedures described previously [39].

The closest sequence in the database (https://www.ddbj.nig.ac.jp/) was determined by a BLAST [40] search, and multiple sequence alignments were constructed using ClustalW 2.1 [41]. Alignments were manually edited, and phylogenetic trees with the related reference genes were constructed using ClustalW 2.1 with the neighbor-joining method.

*2.4. Analysis of the Community Structure of Endophytes*

Based on the results of the BLAST search and phylogenetic analysis, relative abundance (%) was calculated according to the class and genus of the identified bacteria for each sample, reflecting the relative abundance on the plate (Table 1). These results were used to analyze the effects of OYK inoculation, the difference between the presence and absence of a nitrogen source in the medium, and the two sweet potato cultivars on the community structure of the endophytes. Principal component analysis (PCA) was applied on a genera basis using IBM SPSS Statistics ver. 25 (IBM Co., Armonk, NY, USA).

*2.5. Nucleotide Sequence Accession Numbers*

The sequence data generated in this study were deposited in the DDBJ Nucleotide Submission System under the accession numbers LC583148 to LC583248.

*2.6. Statistical Analysis*

Statistical analysis of the sweet potato cultivation data was performed using Student's *t*-test. The Shannon diversity index (*H'*) was calculated based on the identified genus to characterize the diversities in the endophytic bacterial communities.

## 3. Results

*3.1. Effects of OYK Inoculation*

In terms of the dry weights of shoots and tubers, the growth of sweet potato cultivar Beniharuka was better than that of Beniazuma, and there was no significant difference between samples with and without OYK inoculation in either cultivar (Figure 1).

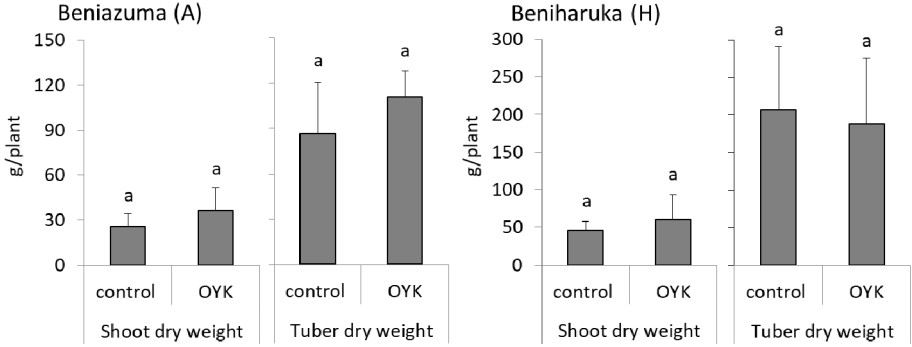

**Figure 1.** Dry weight of two sweet potato cultivars, Beniazuma (A) and Beniharuka (H), inoculated with OYK as PGPR, compared with the control. The bars represent standard deviation (n = 3), and different letters indicate significant differences at *p* < 0.05 by Student's *t*-test.

*3.2. Isolation of Endophytic Bacterial Strains*

Originally, 269 bacterial colonies appeared on the agar plates in total, of which 232 strains were successfully isolated. On the basis of their observed morphologies on the modified MR agar medium, with and without nitrogen supplementation, the isolates were grouped into 6–17 groups in each sample. Based on their relative abundance, 1–3 representative isolates were selected from each group, comprising 30–81% of the original isolates; as a result, 109 isolates were selected, in total, for further analysis (Table 1).

### 3.3. Genetic Analysis of Endophytes

Among the 109 selected endophytic bacterial isolates, 101 strains were successfully sequenced for the partial 16S rRNA gene. The results of the closest relatives in the DDBJ database are presented in Supplementary materials Table S3 and Figure S1 and summarized in Table 2 and Figure 2. The isolates belonged to 25 bacterial genera in 9 classes, which showed 97–100% homology. Among the 101 identified bacterial strains, 55 representative strains from each genus in each sample were subjected to PCR for the *nifH* gene; however, none of the strains produced positive amplification, with *Bradyrhizobium elkanii* USDA 94 used as a positive control.

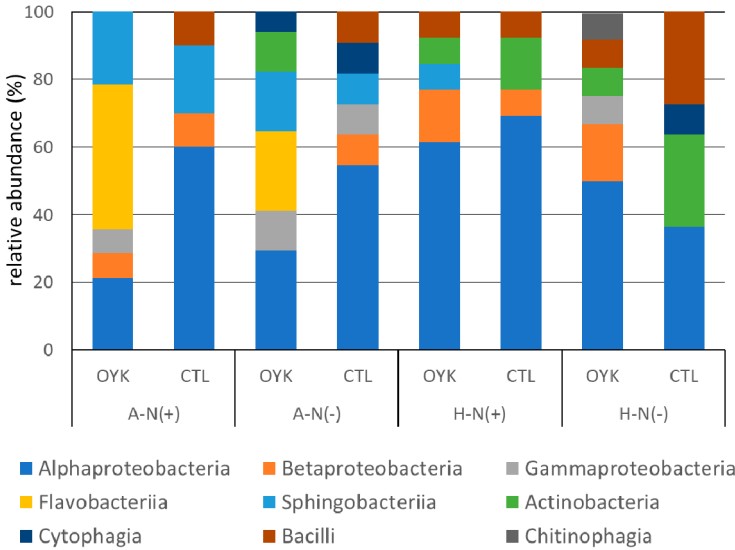

**Figure 2.** Relative class composition of endophytes of two sweet potato cultivars, Beniazuma (A) and Beniharuka (H), inoculated with OYK as PGPR, compared with the control. Bacteria were cultured using a modified MR medium, with and without a supplemental nitrogen source.

### 3.4. Community Structure of Endophytes

In control samples, α-Proteobacteria predominated (36–69%) in both cultivars, in which *Novosphingobium* sp. was dominant (36–54%). After the inoculation of OYK, the fate of *Novosphingobium* sp. was different between the cultivars. In Beniazuma, *Novosphingobium* sp. disappeared, while it remained (25–38%) in Beniharuka. *Rhizobium* sp. in N(+) disappeared in both cultivars after inoculation. With the disappearance of or decrease in *Novosphingobium* sp. and *Rhizobium* sp., two other genera in α-Proteobacteria, *Sphingomonas* sp. (6–21%) and *Sphingobium* sp. (8–15%), newly appeared, and *Chryseobacterium* sp. (21–24%) and *Acinetobacter* sp. (21%) in Flavobacteriia also appeared in Beniazuma. Bacilli (8–10%) disappeared only in Beniazuma after inoculation, while it persisted in Beniharuka. While Sphingobacteriia tended to be detected in Beniazuma (9–21%), Actinobacteria was detected in Beniharuka (8–27%), and β-Proteobacteria was similarly detected in both cultivars (7–17%).

To further elucidate the influence of the OYK inoculation, PCA was conducted to evaluate the relative abundance of the endophytic genera in Table 2. The first and second component factors explained 61.1% and 13.8% of the variation, respectively (Figure 3). All control samples, including both cultivars and both media conditions, were positioned close to each other, while the OYK-inoculated samples were positioned farther apart for each cultivar, especially in Beniazuma. The effects of the presence or absence of nitrogen in the media were not apparent.

**Table 2.** Relative abundance (%) of endophytes from two cultivars of sweet potato, with and without OYK inoculation as PGPR. Bacteria were cultured using a modified MR medium, with and without a supplemental nitrogen source.

| Class/Genus | Beniazuma (A) | | | | Beniharuka (H) | | | |
|---|---|---|---|---|---|---|---|---|
| | N (+) | | N (−) | | N (+) | | N (−) | |
| | OYK | CTL | OYK | CTL | OYK | CTL | OYK | CTL |
| **α-Proteobacteria** | **21** | **60** | **29** | **55** | **62** | **69** | **50** | **36** |
| Novosphingobium | - | 50 | - | 45 | 38 | 54 | 25 | 36 |
| Rhizobium | - | 10 | 6 | - | - | 15 | - | - |
| Sphingomonas | 21 | - | 6 | - | 8 | - | 8 | - |
| Sphingobium | - | - | 12 | - | 15 | - | 8 | - |
| Caulobacter | - | - | 6 | 9 | - | - | 8 | - |
| **β-Proteobacteria** | **7** | **10** | **-** | **9** | **15** | **8** | **17** | **-** |
| Methylibium | - | 10 | - | - | - | - | - | - |
| Burkholderia | - | - | - | 9 | - | - | - | - |
| Variovorax | 7 | - | - | - | 8 | 8 | - | - |
| Mitsuaria | | - | - | - | 8 | - | 17 | |
| **γ-Proteobacteria** | **7** | **-** | **12** | **9** | **-** | **-** | **8** | **-** |
| Pseudoxanthomonas | 7 | - | - | - | - | - | 8 | - |
| Stenotrophomonas | - | - | 6 | - | - | - | - | - |
| Pseudomonas | - | - | 6 | - | - | - | - | - |
| Dyella | - | - | - | 9 | - | - | - | - |
| **Flavobacteria** | **43** | **-** | **24** | **-** | **-** | **-** | **-** | **-** |
| Chryseobacterium | 21 | - | 24 | - | - | - | - | - |
| Acinetobacter | 21 | - | - | - | - | - | - | - |
| **Sphingobacteria** | **21** | **20** | **18** | **9** | **8** | **-** | **-** | **-** |
| Mucilaginibacter | - | 20 | - | 9 | 8 | - | - | - |
| Sphingobacterium | 21 | - | - | - | - | - | - | - |
| Pedobacter | - | - | 18 | - | - | - | - | - |
| **Actinobacteria** | **-** | **-** | **12** | **-** | **8** | **15** | **8** | **27** |
| Microbacterium | - | - | - | - | 8 | 15 | 8 | 27 |
| Streptomyces | - | - | 6 | - | - | - | - | - |
| Lysinimonas | - | - | 6 | - | - | - | - | - |
| **Cytophagia** | **-** | **-** | **6** | **9** | **-** | **-** | **-** | **9** |
| Dyadobacter | - | - | 6 | - | - | - | - | 9 |
| Chryseolinea | - | - | - | 9 | - | - | - | - |
| **Bacilli** | **-** | **10** | **-** | **9** | **8** | **8** | **8** | **27** |
| Bacillus | - | 10 | - | 9 | 8 | 8 | 8 | 27 |
| **Chitinophagia** | **-** | **-** | **-** | **-** | **-** | **-** | **8** | **-** |
| Filimonas | - | - | - | - | - | - | 8 | - |

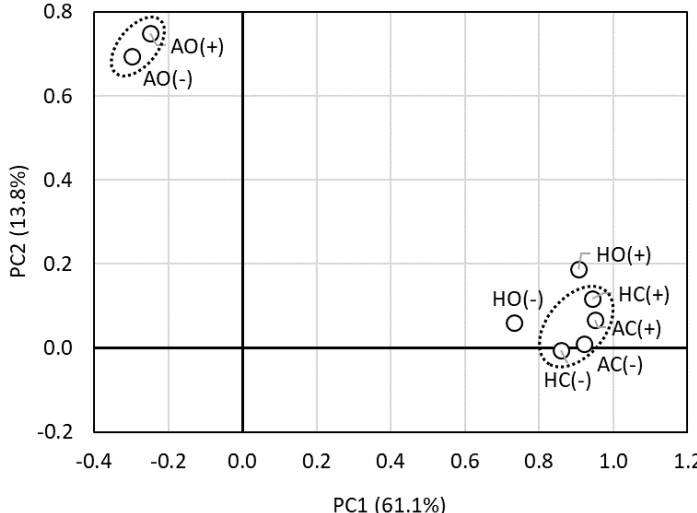

**Figure 3.** Principal component analysis (PCA) of endophytic communities of two sweet potato cultivars, Beniazuma (A) and Beniharuka (H), inoculated with OYK (O) as PGPR, compared with the control (C). Bacteria were cultured using a modified MR medium, with (+) and without (−) a supplemental nitrogen source. PCA was performed based on the bacterial genera in Table 2.

### 3.5. Diversity of Endophytes

Shannon diversity indices (*H*), calculated on the genus level, were increased with the inoculation of OYK in all conditions (Figure 4 and Supplementary materials Figure S2). The increase appeared to be larger in endophytic communities that were isolated using nitrogen-free media, although the indices were similar among the control samples. No difference between the cultivars was apparent.

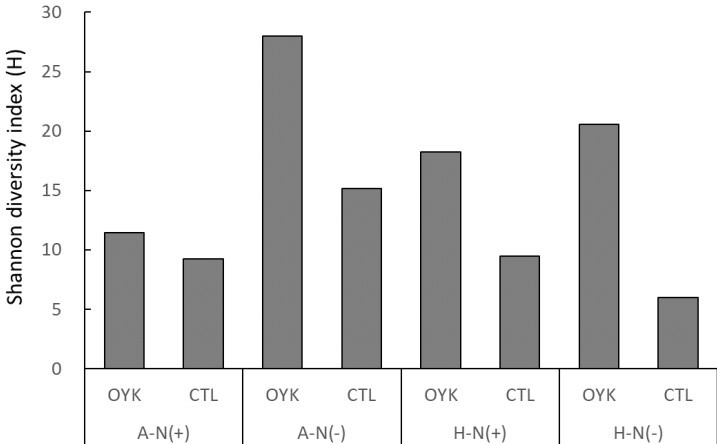

**Figure 4.** Shannon diversity index (H) of endophytic communities of two sweet potato cultivars, Beniazuma (A) and Beniharuka (H), inoculated with OYK as PGPR, compared with the control. Bacteria were cultured using a modified MR medium, with and without a supplemental nitrogen source.

## 4. Discussion

Bacillus strains have been well recognized as PGPR for their plant-growth-promoting performance in sweet potato [8], tomato [42–45], mulberry [46], lettuce [47], wheat [48], pepper [49], potato [50], tobacco [6,51], and saffron [52], as well as their antimicrobial activities against pathogens [27,28], and they are commercially available for their potential use in agriculture [53,54]. However, in our study, the PGPR properties of OYK were not observed (Figure 1). One possible reason might be that the

inoculated OYK disappeared during the cultivation due to environmental factors and competition with indigenous rhizobacteria, as discussed below.

The endophytic community structure has been reported to be determined by several factors, such as plant genotype, soil type [55], and environmental conditions, as well as stochastic sampling factors [56]. In the present study, analysis of the bacterial endophytes of sweet potato revealed that Proteobacteria was the dominant phylum in the communities, followed by Flavobacteria, Sphingobacteria, Actinobacteria, and Bacilli. α-Proteobacteria was the dominant class in Proteobacteria, followed by β- and γ-Proteobacteria (Table 2). In previous studies of sweet potato endophytes, Proteobacteria, including α-, β-, and γ-Proteobacteria, Flavobacteria, Actinobacteria, and Bacilli were also predominant among isolates [27,28,57]. These results suggest that the endophytic community of sweet potato consists of bacteria belonging to common phyla.

Almost all of the detected genera in Proteobacteria, Actinobacteria, and Bacilli have been reported as endophytes in sweet potato [27,28,57] except for *Novosphingobium* sp., which was the dominant genus in most samples. The other dominant genera in our study, *Chryseobacterium* sp., *Acinetobacter* sp., *Mucilaginibacter* sp., and *Sphingobacterium* sp., have not been reported as endophytes. The genera in Flavobacteria and Sphingobacteria were isolated from the cultivar Beniazuma, suggesting that these isolates were sweet potato cultivar-dependent. Differences in endophytic and rhizosphere bacterial communities among sweet potato cultivars have also been demonstrated [27,58]. On the other hand, the common dominant genera in the other studies, *Enterobacter* sp., *Pantoea* sp., *Luteibacter* sp., *Herbaspirillum* sp., and *Curtobacterium* sp., were not isolated in our study, suggesting the presence of diverse bacterial endophytes of sweet potato, with some common genera.

The inoculation of OYK changed the composition of the indigenous bacterial endophytic communities on both the phylum and genus levels, though OYK itself failed to maintain a population as an endophyte. The effects were similar between N(+) and N(−) media, while they were different between the Beniazuma (A) and Beniharuka (H) cultivars, especially for *Novosphingobium* sp., which was dominant in all control samples and disappeared in Beniazuma (A) while remaining predominant in Beniharuka (H). Flavobacteria and Sphingobavteria in Beniazuma (A) only appeared after the inoculation of OYK, which could have caused the change in the community structures found in PCA (Figure 3). Although only one sample of the sweet tuber was used for each cultivar and media condition, the closer positions of the control samples indicate that variability in the community structures of the control samples was within a certain range and that the different positions in PCA were caused by the inoculation of OYK. These results suggest that interactive endophytic bacterial behavior might be influenced by the cultivar of sweet potato. It has been reported that the plant cultivar and genotype affect communities of rhizobacteria, presumably as a result of competition for different root exudates [59–61]. Differences in a rhizobacterial community might affect the corresponding endophytic community as a result. Germida et al. [62] compared rhizoplane and endophytic bacteria strains that were isolated from canola plants and suggested that endophytes are a subset of the rhizoplane community. Additionally, differences in nutritional compositions of endophytic environments will also affect the community through competition.

In a seed and soil inoculation experiment with *Bacillus* spp., the Bacillus inocula failed to establish as endophytes in broccoli roots, as in our study, and the main effects of the Bacillus inoculation were a reduction in *Lysobacter* and *Acidovorax* and an increase in *Acinetobacter*, as analyzed by metagenomic sequencing [31]. The authors also reported that the addition of *B. amyloliquefaciens* influenced the endophytic microbial community: the most common *Pseudomonas* endophytes decreased in abundance, accompanied by an increase in *Dyadobacter*, *Variovorax*, *Tahibacter*, and *Sphingomonas*. In contrast, the inoculation of *B. cereus* and *B. subtilis* did not affect the population of *Pseudomonas* though it changed the endophytic community composition of minor genera. Although the genera affected by the Bacillus inoculation were different from those in our study, the results obtained by culture-dependent and -independent studies suggest that a microbial inoculation can change an endophytic microbial community, even if the inoculant cannot establish a population as an endophyte. As many studies

have shown the importance of endophytes for plant growth promotion, elucidating the interaction mechanisms is an essential line of research.

Although *Bacillus* spp. have been reported as indigenous endophytes in sweet potato [28,57,58] and in other crops such as tomato [63], banana [64], canola [62], and switchgrass [65], the inoculated OYK and *Bacillus* spp. strains [31] could not establish populations as endophytes. On the other hand, the inoculation of endophytic *Bacillus subtilis,* isolated from wheat, could establish a population in wheat root and showed potential as a biological control against plant pathogens [66]. Changes in the compositions of plant metabolites and root exudates that would be caused by OYK might directly change indigenous rhizospheric and endophytic microbial communities and/or might indirectly prevent the successful colonization of OKY due to competition with microbial communities for compounds. As OYK was isolated from the soil, the endophytic potential of an inoculant, whether it was originally isolated as an endophyte, seems to be important.

The Shannon diversity index (*H*) of the isolated endophytic community increased with OYK inoculation (Figure 3). The tendency was the same as that in the results obtained by Gadhave et al. [31], who also reported an increase in diversity in both *Bacillus amyloliquefaciens*- and mixed *Bacillus* spp.-treated sprouting broccoli, examined by a culture-independent metagenomic approach. In both studies, using different approaches, the number of genera identified increased with the inoculation; however, the mechanisms are still unclear.

**Supplementary Materials:** The following are available online at http://www.mdpi.com/2311-7524/6/4/81/s1, Figure S1: Phylogenetic tree of endophytes of sweet potato cultivars, Beniazuma (A) and Beniharuka (H), inoculated with PGPR, OYK compared with control, using modified MR medium supplemented with and without nitrogen source based on partial 16S rRNA gene sequences, Figure S2: Relative genus composition of endophytes of sweet potato cultivars, Beniazuma (A) and Beniharuka (H), inoculated with PGPR, OYK compared with control, using modified MR medium supplemented with and without nitrogen source, Table S1: Ingredients of modified MR (N-free MR) agar medium, Table S2: PCR ingredients for amplification of 16S rRNA and nifH genes, Table S3: Closest relatives of endophytic bacterial strains from two cultivars of sweet potato inoculated with and without PGPR, OYK, using modified MR medium supplemented with and without nitrogen source.

**Author Contributions:** A.S. and K.I. conceptualized the study and designed the experiments; A.S. performed the experiments; the field experiment was performed by F.A. and S.H.; M.H.R.H. assisted in the experiment, data recording, and analysis; A.S. wrote the article, with a substantial contribution from K.I. All authors have read and agreed to the published version of the manuscript.

**Funding:** This research received no external funding.

**Conflicts of Interest:** The authors declare no conflict of interest.

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
