# Peer review of "Effects of the Biofertilizer OYK (Bacillus sp.) Inoculation on Endophytic Microbial Community in Sweet Potato"

_horticulturae, doi:10.3390/horticulturae6040081_

Round 1
Reviewer 1 Report
Comments to manuscript ID: horticulturae-965666: Effects of the biofertilizer OYK (Bacillus sp.) inoculation on endophytic microbial community in sweet potato, submitted by Salehin et al.
The manuscript addresses a very interesting field by investigating the influence of the commercial biofertilizer 17 OYK with Bacillus sp. on the endophytic bacterial communities of two cultivars of sweet potato, Beniazuma and Beniharuka, cultivated under experimental field conditions. The differences reported are well worked out by adequate methods and properly conducted experiments, but the findings stay nevertheless descriptive. The authors concluded from their results, in line with published ones of other groups, that the application of PGRP bacteria can change plant endophytic microbial communities, which is rather important for agricultural use of inoculants. However, the consequences of the changes are still enigmatic.
There are several concerns which have to be considered.
1) The authors do not discuss sufficiently why the OKY Bacillus strain may disappear during cultivation. What for environmental and competition factors could be responsible for the disappearance? Are there differences between the two cultivars regarding exuded secondary products and primary metabolites or are biosynthesis of secondary products and compositions of root exudates changed as a response to inoculation, preventing a successful colonization of OKY, thereby changing the established endophytic microbial community?
2) The results of the experiments with and without nitrogen source remains elusive. Give a better explanation.
3) Line 35: A forecast is an event that might occur or might not occur. The sentence is misleading. Change wording.
4) A major concern: The language must be thoroughly corrected by a native speaker. There are many incomplete sentences and wrong wordings. Several sentences are therefor difficult to understand.
Author Response
Thank you for your critical review on our manuscript. We revised it according to your suggestions and English editing by MDPI. Our a point-by-point response to the reviewer’s comments are as follows:
There are several concerns which have to be considered.
1) The authors do not discuss sufficiently why the OKY Bacillus strain may disappear during cultivation. What for environmental and competition factors could be responsible for the disappearance? Are there differences between the two cultivars regarding exuded secondary products and primary metabolites or are biosynthesis of secondary products and compositions of root exudates changed as a response to inoculation, preventing a successful colonization of OKY, thereby changing the established endophytic microbial community?
The possible reasons for the disappearance of OYK were discussed in the paragraph below and revised according to the reviewer’s suggestion as follows:
Changes in the compositions of plant metabolites and root exudates that would be caused by OYK might directly change indigenous rhizospheric and endophytic microbial communities and/or might indirectly prevent the successful colonization of OKY due to competition with microbial communities for compounds. (319-323)
2) The results of the experiments with and without nitrogen source remains elusive. Give a better explanation.
By further analysis using principal component analysis (PCA) based on the composition of the endophytic genera in each sample, difference of the community structures affected by OYK inoculation, cultivars and media conditions was clearly presented. (126-128, 231-236)
3) Line 35: A forecast is an event that might occur or might not occur. The sentence is misleading. Change wording.
We revised the sentence according to the reviewer’s suggestion as follows:
Modern agriculture systems are being intensified through the use of various technologies to achieve maximum efficiency and highly qualified products to meet the growing global demand for food supply. (35-37)
4) A major concern: The language must be thoroughly corrected by a native speaker. There are many incomplete sentences and wrong wordings. Several sentences are therefor difficult to understand.
We revised our manuscript according to English editing by MDPI.
Reviewer 2 Report
Overall the topic of the study is relevant. However, I have some doubts and questions about experimental procedures.
Comments:
- Page 2. lane 83-84. There should be provided more information about OYK biofertilizer. Is it composed only of one Bacillus sp. strain? Or there are several different Bacillus strains? Is the species of Bacillus sp. strain known? What is CFU of Bacillus sp. in biofertilizer? Are there vegetative cells or endospores?
- Page 2 lane 85. Biofertilizer was diluted with distilled water. Was the distilled water sterile? Why biofertilizer was diluted in distilled water? Didn't incubation of Bacillus sp. cells in distilled water for 60 h have some impact on their viability?
- Page 3 lane 96-97. "Surface of the tuber samples were washed with running tap water". Tap water isn't sterile, so by washing potatoes with tap water you contaminate them with microorganisms residing in the water supply system. Then the tuber was cut, but was the knife sterilised before cutting the tuber?
- Page 3 lane 102. Isolates, but not strains were selected.
- Table 1 is not entirely clear. Instead of "Original" you should use the term CFU (colony forming units), I think. Secondly, you have isolated different isolates and you didn't test if all these isolates belong to different strains. Maybe some of the isolates belong to the same strain? So you should use the term isolate instead of strain.
- Page 4 lane 161 it would good to explain why you performed PCR for nifH gene?
- The sample size is to low to make a conclusion about the structure of the community of endophytes. And what would the results if you compared the diversity of endophytes between two different potatoes thumbs belonging to the same cultivar and subjected to the same growth conditions? Maybe each separate potatoes tuber had different microbiota before the experiment?
- Table 2. Some formating errors.
Author Response
Thank you for your critical review on our manuscript. We revised it according to your suggestions and English editing by MDPI. Our a point-by-point response to the reviewer’s comments are as follows:
1) Page 2. lane 83-84. There should be provided more information about OYK biofertilizer. Is it composed only of one Bacillus sp. strain? Or there are several different Bacillus strains? Is the species of Bacillus sp. strain known? What is CFU of Bacillus sp. in biofertilizer? Are there vegetative cells or endospores?
OYK is one Bacillus sp. strain which showed the most similar 16S rRNA gene sequence with Bacillus cereus MH19 with about 8E+9 CFU/ml in the state of endospores. (85)
2) Page 2 lane 85. Biofertilizer was diluted with distilled water. Was the distilled water sterile? Why biofertilizer was diluted in distilled water? Didn't incubation of Bacillus sp. cells in distilled water for 60 h have some impact on their viability?
Sterilized distilled water was used in dilution. The procedures were according to the manufacturer's instruction as described in the manuscript. (86-87) We did not examine their viability during the inoculation.
3) Page 3 lane 96-97. "Surface of the tuber samples were washed with running tap water". Tap water isn't sterile, so by washing potatoes with tap water you contaminate them with microorganisms residing in the water supply system. Then the tuber was cut, but was the knife sterilized before cutting the tuber?
Tap water was wiped off with a paper towel after washing, then efficiency of the washing procedure was evaluated by toughing the surface of the tuber on to the MR media. The knife was sterilized before use. (98-102)
4) Page 3 lane 102. Isolates, but not strains were selected.
We corrected the word according to the reviewer’s suggestion. (105)
5) Table 1 is not entirely clear. Instead of "Original" you should use the term CFU (colony forming units), I think. Secondly, you have isolated different isolates and you didn't test if all these isolates belong to different strains. Maybe some of the isolates belong to the same strain? So you should use the term isolate instead of strain.
We revised Table 1 according to the reviewer’s suggestion, and the correction was applied to the other part of the manuscript.
6) Page 4 lane 161 it would good to explain why you performed PCR for nifH gene?
Explanation was described in Materials and Methods. (110-112)
7) The sample size is too low to make a conclusion about the structure of the community of endophytes. And what would the results if you compared the diversity of endophytes between two different potatoes thumbs belonging to the same cultivar and subjected to the same growth conditions? Maybe each separate potatoes tuber had different microbiota before the experiment?
As the reviewer suggested, the community structures could not be the same among tubers in the same plant. Then we additionally compared the community structures of the samples by PCA analysis. The results (Figure 3) indicated that all control samples, including both cultivars and both media conditions, were positioned close to each other, while the OYK-inoculated samples were positioned father apart for each cultivar, especially in Beniazuma, suggesting that variability in the community structures of the control samples was within a certain range, and that the different positions in PCA were caused by the inoculation of OYK. (125-127, 230-235, 288-292)
8) Table 2. Some formating errors.
The format of Table 2 was corrected.
Round 2
Reviewer 2 Report
Thank the authors for their response. In my opinion, the revised manuscript is suitable for publication.
Author Response
Thank you for your comments.